# Inhibition of osteoblastic Smurf1 promotes bone formation in mouse models of distinctive age-related osteoporosis

Chao Liang [1,2,3], Songlin Peng[1,4], Jie Li[5,6], Jun Lu[1,2,3], Daogang Guan[1,2,3], Feng Jiang[1,7], Cheng Lu[1,2,3,8], Fangfei Li[1,2,3], Xiaojuan He[1,2,3,8], Hailong Zhu[2], D.W.T. Au[9], Dazhi Yang[4], Bao-Ting Zhang[5], Aiping Lu[1,2,3,10] & Ge Zhang[1,2,3]

Bone morphogenetic protein (BMP) signaling is essential for osteogenesis. However, recombinant human BMPs (rhBMPs) exhibit large inter-individual variations in local bone formation during clinical spinal fusion. Smurf1 ubiquitinates BMP downstream molecules for degradation. Here, we classify age-related osteoporosis based on distinct intraosseous BMP-2 levels and Smurf1 activity. One major subgroup with a normal BMP-2 level and elevated Smurf1 activity (BMP-2$^n$/Smurf1$^e$) shows poor response to rhBMP-2 during spinal fusion, when compared to another major subgroup with a decreased BMP-2 level and normal Smurf1 activity (BMP-2$^d$/Smurf1$^n$). We screen a chalcone derivative, i.e., 2-(4-cinnamoylphenoxy) acetic acid, which effectively inhibits Smurf1 activity and increases BMP signaling. For BMP-2$^n$/Smurf1$^e$ mice, the chalcone derivative enhances local bone formation during spinal fusion. After conjugating to an osteoblast-targeting and penetrating oligopeptide (DSS)$_6$, the chalcone derivative promotes systemic bone formation in BMP-2$^n$/Smurf1$^e$ mice. This study demonstrates a precision medicine-based bone anabolic strategy for age-related osteoporosis.

[1] Law Sau Fai Institute for Advancing Translational Medicine in Bone and Joint Diseases, School of Chinese Medicine, Hong Kong Baptist University, 999077 Hong Kong, SAR, China. [2] Institute of Integrated Bioinfomedicine and Translational Science, School of Chinese Medicine, Hong Kong Baptist University, 999077 Hong Kong, SAR, China. [3] Institute of Precision Medicine and Innovative Drug Discovery, HKBU Institute for Research and Continuing Education, 518000 Shenzhen, China. [4] Department of Spine Surgery, Shenzhen People's Hospital, Ji Nan University Second College of Medicine, 518020 Shenzhen, China. [5] School of Chinese Medicine, Faculty of Medicine, Chinese University of Hong Kong, 999077 Hong Kong, SAR, China. [6] Clinical Medical Laboratory of Peking University Shenzhen Hospital, 518036 Shenzhen, China. [7] Zhejiang Pharmaceutical College, 315100 Ningbo, China. [8] Institute of Basic Research in Clinical Medicine, China Academy of Chinese Medical Sciences, 100700 Beijing, China. [9] Department of Biology and Chemistry, City University of Hong Kong, 999077 Hong Kong, SAR, China. [10] Institute of Arthritis Research, Shanghai Academy of Chinese Medical Sciences, 200032 Shanghai, China. These authors contributed equally: Chao Liang, Songlin Peng, Jie Li, Jun Lu, Daogang Guan, Feng Jiang. Correspondence and requests for materials should be addressed to B.-T.Z. (email: zhangbaoting@cuhk.edu.hk) or to A.L. (email: aipinglu@hkbu.edu.hk) or to G.Z. (email: zhangge@hkbu.edu.hk)

Bone is a dynamic tissue, which undergoes continuous remodeling throughout life. Bone homeostasis is maintained by a balance between osteoclastic bone resorption and osteoblastic bone formation[1]. However, the homeostasis shifts out of balance during aging, resulting in decreased bone formation relative to bone resorption and age-related osteoporosis[2,3].

Bone morphogenetic protein (BMP) signaling plays a significant role in osteoblastic bone formation[4]. BMPs canonically signal through intracellular transducers (Smad1/5/8). Briefly, BMPs initiate the signaling by binding to cell surface receptors and phosphorylate Smad1/5/8. Phosphorylated Smad1/5/8 (p-Smad1/5/8) associate with Smad4 and translocate into the nucleus, where they transcribe and interact with Runx2 to induce osteogenic gene expression (such as osteocalcin)[5–7]. After the signaling is transferred, Smad1/5/8 are ubiquitinated for degradation by Smad ubiquitination regulatory factor-1 (Smurf1), to prevent the overwhelming BMP signaling activation[7,8]. Moreover, Smurf1 also directly mediates Runx2 degradation in a ubiquitination−dependent manner and thus acts as a bone formation suppressor[9–13].

As initiators of the signaling cascade, BMPs have been considered as powerful bone anabolic agents[14,15]. To date, recombinant human BMPs (rhBMPs) have been clinically approved to promote local bone formation in non-union, fracture repair, and spinal fusion[16,17]. However, emerging evidence demonstrates large inter-individual variations in local bone anabolic potential of rhBMPs[17–20]. Even though possible limitations including rapid clearance of rhBMPs in tissues, may partly account for the less impressive behavior of rhBMPs[18], the underlying explanation for the large inter-individual variations is still poorly understood.

In this study, we find that age-related osteoporotic individuals could be classified into different subgroups based on distinct intraosseous BMP-2 levels and Smurf1 activity. One major subgroup has a normal BMP-2 level and elevated Smurf1 activity (BMP-2$^{normal}$/Smurf1$^{elevated}$, BMP-2$^n$/Smurf1$^e$), whereas another major subgroup demonstrates a decreased BMP-2 level and normal Smurf1 activity (BMP-2$^{decreased}$/Smurf1$^{normal}$, BMP-2$^d$/Smurf1$^n$). Both subgroups show obviously reduced levels of intraosseous p-Smad1 and Runx2 activation, but with different reduction extents. Serum osteocalcin with a consistent reduction pattern is confirmed as a biomarker in stratifying the two subgroups. The BMP-2$^n$/Smurf1$^e$ subgroup shows poor local rhBMP-2 response during spinal fusion when compared to the BMP-2$^d$/Smurf1$^n$ subgroup.

We hypothesize that osteoblastic Smurf1 inhibition could be a precision medicine strategy to promote bone formation in BMP-2$^n$/Smurf1$^e$ subgroup. We design in silico strategy to screen small molecular inhibitors targeting Smurf1. By molecular docking, we identify a chalcone derivative, i.e., 2-(4-cinnamoyl-phenoxy)acetic acid, which significantly inhibits Smurf1 activity, improves BMP signaling and osteogenic differentiation and promotes local bone formation during spinal fusion for BMP-2$^n$/Smurf1$^e$ subgroup of osteoporotic mice.

Previously, we identified an oligopeptide (AspSerSer)$_6$ ((DSS)$_6$), which could specifically recognize bone formation surfaces and showed great potential as a targeting moiety for osteoblasts[21]. Further study demonstrates that (DSS)$_6$ had a cell-penetrating property[22]. We conjugate the chalcone derivative to (DSS)$_6$ and find that (DSS)$_6$ facilitates the chalcone derivative targeting osteoblasts and inhibiting Smurf1 activity, leading to significantly promoted systemic bone formation in BMP-2$^n$/Smurf1$^e$ subgroup of osteoporotic mice. All these results indicate that osteoblastic Smurf1 inhibition could be a precision medicine-based bone anabolic strategy for BMP-2$^n$/Smurf1$^e$ subgroup of age-related osteoporotic individuals.

## Results

**Subgroup classification of aged osteoporotic patients.** We collected bone specimens and blood from aged osteoporotic vertebral compression fracture (VCF) patients (60–69 and 70–79 years old) and adult traumatic VCF patients (30–39 years old) (Supplementary Table 1). Compared to adult traumatic VCF patients, aged osteoporotic VCF patients showed a decreased intraosseous BMP-2 level with a large variation (Fig. 1a). However, elevated Smurf1 activity (Smad1 bound to Smurf1) with a large variation but unaltered Smurf1 level was observed in aged patients (Fig. 1a and Supplementary Fig. 1a). Based on distinct intraosseous BMP-2 levels and Smurf1 activity (Smad1 bound to Smurf1), aged patients were classified into subgroups, in which one major subgroup (BMP-2$^{normal}$/Smurf1$^{elevated}$, BMP-2$^n$/Smurf1$^e$) had a normal BMP-2 level and elevated Smurf1 activity, whereas another major subgroup (BMP-2$^{decreased}$/Smurf1$^{normal}$, BMP-2$^d$/Smurf1$^n$) demonstrated a decreased BMP-2 level and normal Smurf1 activity (Fig. 1b). To determine whether sex difference was associated with the above findings, we compared the levels of intraosseous BMP-2 and Smurf1 activity between male and female osteoporotic VCF patients aged 60–69 or 70–79 years old. Our results showed that the male and female patients had comparable intraosseous levels of BMP-2 and Smurf1 activity in each age category (Supplementary Fig. 1b). Both male and female patients exhibited subgroup classification including a BMP-2$^n$/Smurf1$^e$ subgroup and BMP-2$^d$/Smurf1$^n$ subgroup (Supplementary Fig. 1c), indicating that the subgroup classification was sex-independent. Furthermore, elevation of Smurf1 activity (Smad1 bound to Smurf1, ubiquitination of Smad1 and Runx2) in BMP-2$^n$/Smurf1$^e$ subgroup and reduction of BMP-2 in BMP-2$^d$/Smurf1$^n$ subgroup were in age-related manners (Fig. 1c, d, Supplementary Fig. 1d, 1e). Both subgroups showed obviously reduced levels of intraosseous p-Smad1 and Runx2 activation, as well as serum osteocalcin in the above two age categories. The reduction extents were more significant in BMP-2$^d$/Smurf1$^n$ subgroup than BMP-2$^n$/Smurf1$^e$ subgroup (Figs 1e, f, Supplementary Fig. 1f).

**Distinct rhBMP-2 response in osteoporotic patient subgroups.** We collected bone specimens and blood from aged osteoporotic lumbar spinal stenosis (LSS) patients (60–69 years old) and adult lumbar disc herniation (LDH) patients (30–39 years old) (Supplementary Table 2). Consistent with the findings from aged osteoporotic VCF patients, there were comparable Smurf1 expression, but decreased BMP-2 level and elevated Smurf1 activity (Smad1 bound to Smurf1) with large variations in aged osteoporotic LSS patients, when compared to those in adult LDH patients (Fig. 2a and Supplementary Fig. 2a). The aged patients were also classified into different subgroups, including a major BMP-2$^n$/Smurf1$^e$ subgroup with a normal BMP-2 level and elevated Smurf1 activity (Smad1 bound to Smurf1, ubiquitination of Smad1 and Runx2) and another major BMP-2$^d$/Smurf1$^n$ subgroup with a decreased BMP-2 level and normal Smurf1 activity (Fig. 2b and Supplementary Fig. 2b). The reduction extents of intraosseous p-Smad1, Runx2 activation, and serum osteocalcin in BMP-2$^d$/Smurf1$^n$ subgroup was more significant (Fig. 2c, d, Supplementary Fig. 2c). During lumbar decompression and spinal fusion, both subgroups were locally administered with rhBMP-2 (INFUSE® Bone Graft). Computerized tomography (CT) scan showed that BMP-2$^n$/Smurf1$^e$ subgroup had less newly formed bone in L4-L5 or L5-S1 vertebrae, while there was massive bone formation in BMP-2$^d$/Smurf1$^n$ subgroup (Fig. 2e). Spinal fusion incidence and serum osteocalcin level were much lower in BMP-2$^n$/Smurf1$^e$ subgroup when compared to BMP-2$^d$/Smurf1$^n$ subgroup (Fig. 2f, g).

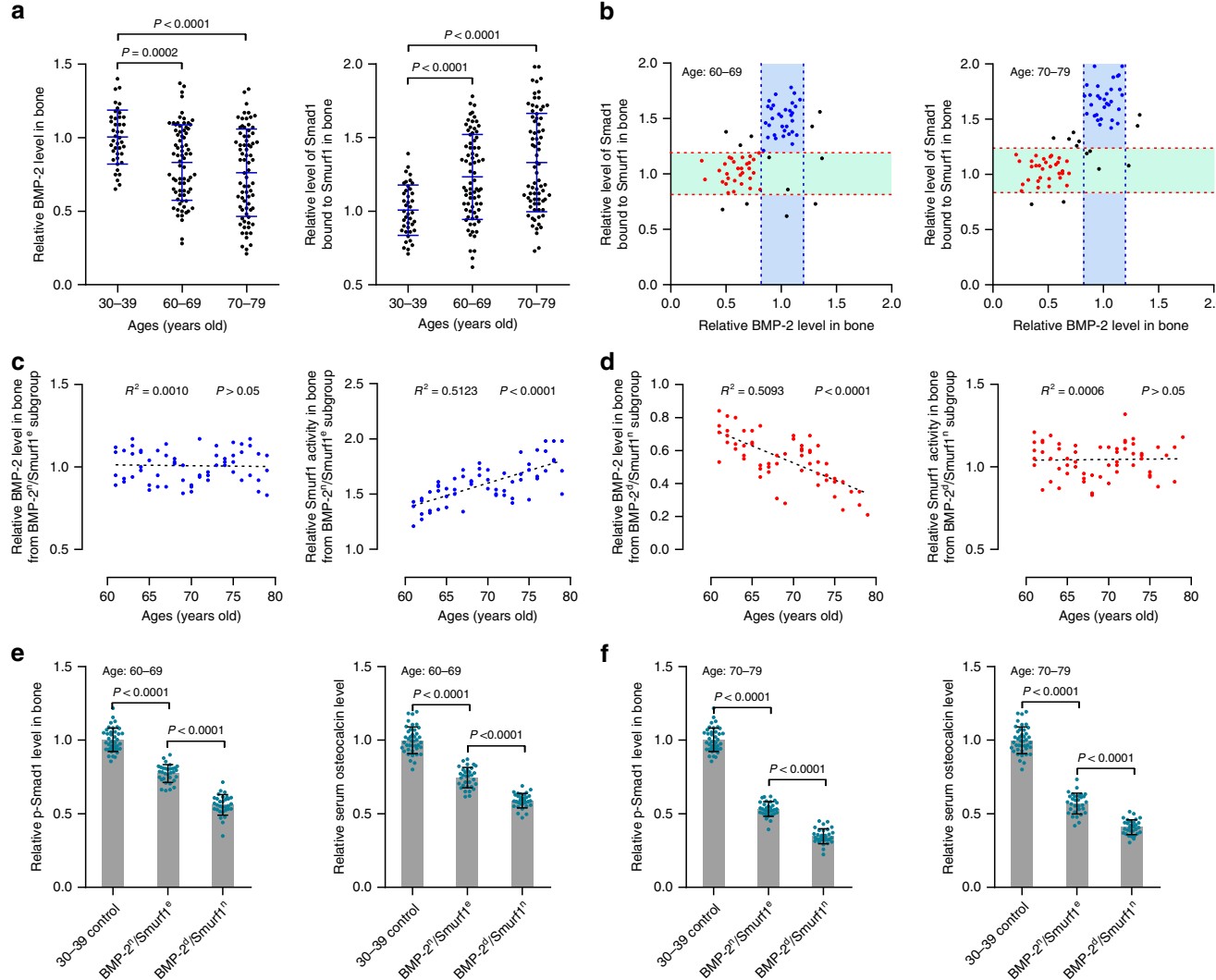

**Fig. 1** BMP signaling, Smurf1 activity and subgroup classification of aged osteoporotic patients. **a** Relative levels of intraosseous BMP-2 (left) and Smurf1 activity (Smad1 bound to Smurf1, right) in osteoporotic VCF patients aged 60–69 ($n = 75$) or 70–79 years old ($n = 75$). **b** Classification of aged patients into subgroups based on intraosseous BMP-2 levels and Smurf1 activity (Smad1 bound to Smurf1) (left, 60–69 years old; right, 70–79 years old). The cluster of blue dots, BMP-2$^n$/Smurf1$^e$ subgroup; the cluster of red dots, BMP-2$^d$/Smurf1$^n$ subgroup. Relative levels of intraosseous BMP-2 (mean ± s.d., indicated by the two blue dashed lines) and Smad1 bound to Smurf1 (mean ± s.d., indicated by the two red dashed lines) in adult traumatic VCF patients (30–39 years old) served as cutoff parameters. **c** Relative levels of intraosseous BMP-2 (left) and Smad1 bound to Smurf1 (right) in BMP-2$^n$/Smurf1$^e$ subgroup ($n = 63$). **d** Relative levels of intraosseous BMP-2 (left) and Smad1 bound to Smurf1 (right) in BMP-2$^d$/Smurf1$^n$ subgroup ($n = 62$). **e** Relative levels of intraosseous p-Smad1 (left) and serum osteocalcin (right) in BMP-2$^n$/Smurf1$^e$ subgroup ($n = 32$) and BMP-2$^d$/Smurf1$^n$ subgroup ($n = 31$) of patients (60–69 years old). **f** Relative levels of intraosseous p-Smad1 (left) and serum osteocalcin (right) in BMP-2$^n$/Smurf1$^e$ subgroup ($n = 31$) and BMP-2$^d$/Smurf1$^n$ subgroup ($n = 31$) of patients (70–79 years old). The levels of BMP-2, p-Smad1, osteocalcin and Smad1 bound to Smurf1 in each osteoporotic VCF patients from 60–69 or 70–79 age group were normalized to the mean values of adult traumatic VCF patients (30–39 years old) ($n = 41$). Data are mean ± s.d. followed by one-way ANOVA with a post-hoc test

**Subgroup classification of aged osteoporotic mice**. Mice were ovariectomized (OVX) to induce age-related osteoporosis. In accordance with the findings from human, osteoporotic mice showed unaltered Smurf1 expression, but decreased BMP-2 level and elevated Smurf1 activity (Smad1 bound to Smurf1) with large variations, when compared to healthy adult 6-month-old mice (Supplementary Fig. 3a). The osteoporotic mice were classified into BMP-2$^n$/Smurf1$^e$ and BMP-2$^d$/Smurf1$^n$ subgroups (Supplementary Fig. 3b). Elevation of Smurf1 activity (Smad1 bound to Smurf1, ubiquitination of Smad1 and Runx2) in BMP-2$^n$/Smurf1$^e$ subgroup and reduction of BMP-2 level in BMP-2$^d$/Smurf1$^n$ subgroup were in age-related manners (Supplementary Fig. 3c and 3d). Compared to BMP-2$^n$/Smurf1$^e$ subgroup, age-related

reduction of intraosseous p-Smad1, Runx2 activation and serum osteocalcin was more significant in BMP-2$^d$/Smurf1$^n$ subgroup (Supplementary Fig. 3e-3g).

**Serum osteocalcin as a subgrouping biomarker**. Compared to healthy adult 6-month-old mice, there was a decreased level of serum osteocalcin with a large variation in aged osteoporotic mice (Supplementary Fig. 4a). Refer to different reduction extents of serum osteocalcin in BMP-2$^n$/Smurf1$^e$ and BMP-2$^d$/Smurf1$^n$ subgroups determined in Supplementary Fig. 3g, we stratified the mice into different subgroups. One major subgroup with a comparable serum osteocalcin level to that in BMP-2$^n$/Smurf1$^e$

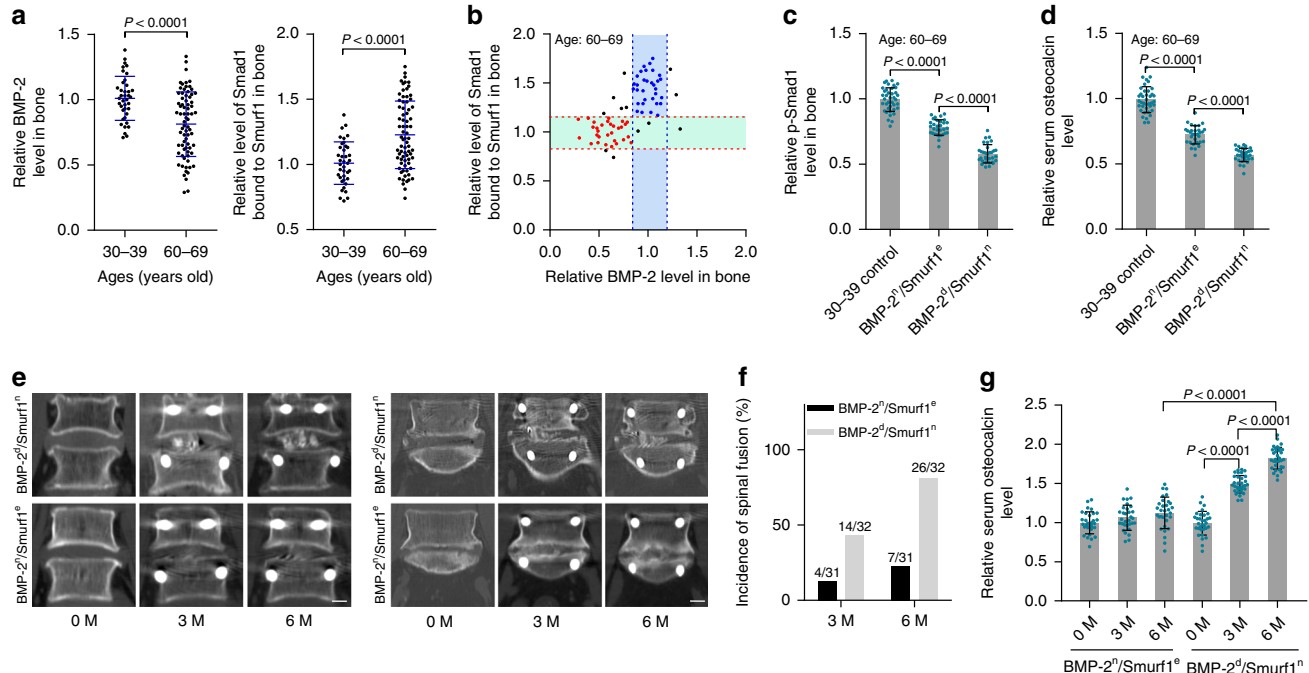

**Fig. 2** Local rhBMP-2 response in human BMP-2$^n$/Smurf1$^e$ and BMP-2$^d$/Smurf1$^n$ subgroups. **a** Relative levels of intraosseous BMP-2 (left) and Smurf1 activity (Smad1 bound to Smurf1, right) in osteoporotic LSS patients aged 60–69 years old ($n = 74$). **b** Classification of osteoporotic LSS patients into subgroups based on intraosseous BMP-2 levels and Smurf1 activity (Smad1 bound to Smurf1). The cluster of blue dots, BMP-2$^n$/Smurf1$^e$ subgroup; the cluster of red dots, BMP-2$^d$/Smurf1$^n$ subgroup. Relative levels of intraosseous BMP-2 (mean ± s.d., indicated by the two blue dashed lines) and Smad1 bound to Smurf1 (mean ± s.d., indicated by the two red dashed lines) in adult LDH patients (30–39 years old) served as cutoff parameters. **c** Relative level of intraosseous p-Smad1 in BMP-2$^n$/Smurf1$^e$ subgroup ($n = 31$) and BMP-2$^d$/Smurf1$^n$ subgroup ($n = 32$). **d** Relative level of serum osteocalcin in the above two subgroups. **e** CT scan showing representative images of spinal fusion in L4-L5 vertebrae (left) or L5-S1 vertebrae (right) in the above two subgroups after local administration of rhBMP-2. Scale bar = 1.0 cm. **f** Incidence of spinal fusion during rhBMP-2 treatment in the above two subgroups, followed by chi-squared test ($P = 0.007$ and $P < 0.0001$ for BMP-2$^d$/Smurf1$^n$ versus BMP-2$^n$/Smurf1$^e$ at 3 M and 6 M, respectively). **g** Relative serum osteocalcin level during rhBMP-2 treatment in the above two subgroups. Prior to rhBMP-2 treatment, the levels of BMP-2, p-Smad1, osteocalcin and Smad1 bound to Smurf1 were normalized to the mean values of adult LDH patients (30–39 years old) ($n = 39$). After the rhBMP-2 treatment for 3 months (3 M) or 6 months (6 M), the levels of osteocalcin in the above two subgroups were normalized to their respective baseline at 0 month (0 M). Data are mean ± s.d. followed by one-way ANOVA with a post-hoc test or Student's $t$ test

subgroup was termed as osteocalcin$^d$ subgroup, while another major subgroup with an equivalent serum osteocalcin level to that in BMP-2$^d$/Smurf1$^n$ subgroup was termed as osteocalcin$^{dd}$ subgroup (Supplementary Fig. 4a). The osteocalcin$^d$ subgroup showed a normal BMP-2 level and elevated Smurf1 activity (Smad1 bound to Smurf1, ubiquitination of Smad1, and Runx2), whereas the osteocalcin$^{dd}$ subgroup demonstrated a decreased BMP-2 level and normal Smurf1 activity (Supplementary Fig. 4b-d). There were obviously reduced levels of intraosseous p-Smad1 and Runx2 activation in both subgroups, but their reduction was more significant in osteocalcin$^{dd}$ subgroup (Supplementary Fig. 4c, d). All these results indicated that serum osteocalcin could be a biomarker in stratifying BMP-2$^n$/Smurf1$^e$ and BMP-2$^d$/Smurf1$^n$ subgroups.

**Distinct rhBMP-2 response in osteoporotic mouse subgroups.** We classified BMP-2$^n$/Smurf1$^e$ and BMP-2$^d$/Smurf1$^n$ subgroups of osteoporotic mice according to serum osteocalcin. The two subgroups of mice received posterolateral intertransverse lumbar fusion surgery in L4-L6 vertebrae and local administration of rhBMP-2 or vehicle (PBS). Compared to BMP-2$^d$/Smurf1$^n$ subgroup, the BMP-2$^n$/Smurf1$^e$ subgroup showed a much lower spinal fusion incidence and serum osteocalcin level (Fig. 3a, b). Radiographic observation by X-ray showed poor spinal fusion in BMP-2$^n$/Smurf1$^e$ subgroup, while there was massive fusion bone

in BMP-2$^d$/Smurf1$^n$ subgroup (Fig. 3c). microCT analysis demonstrated no obvious improvement of fusion bridge, as well as bone mass parameters including bone mineral density (BMD) and relative bone volume (BV/TV) in BMP-2$^n$/Smurf1$^e$ subgroup when compared to those in BMP-2$^d$/Smurf1$^n$ subgroup (Fig. 3d, e). Bone histomorphometric analysis showed no remarkable enhancement of width between fluorochrome labeling bands and bone formation parameters including mineral apposition rate (MAR) and bone formation rate (BFR/BS) in BMP-2$^n$/Smurf1$^e$ subgroup (Fig. 3f, g). In addition, osteoblasts from L4-L6 vertebrae of BMP-2$^n$/Smurf1$^e$ subgroup exhibited no significant increase of p-Smad1 and Runx2 activation when compared to those in osteoblasts from BMP-2$^d$/Smurf1$^n$ subgroup (Supplementary Fig. 5).

**BMP signaling in BMP-2$^n$/Smurf1$^e$ osteoblasts.** Compared to osteoblasts from healthy adult 6-month-old mice, Smurf1 activity (Smad1 bound to Smurf1, ubiquitination of Smad1 and Runx2) was elevated in osteoblasts from BMP-2$^n$/Smurf1$^e$ subgroup but kept unaltered in osteoblasts from BMP-2$^d$/Smurf1$^n$ subgroup of osteoporotic mice. Both subgroups had normal Smurf1 expression (Supplementary Fig. 6a). There were reduced levels of p-Smad1, Runx2 activation and osteocalcin mRNA in osteoblasts from both subgroups, but the reduction was more remarkable in BMP-2$^d$/Smurf1$^n$ subgroup

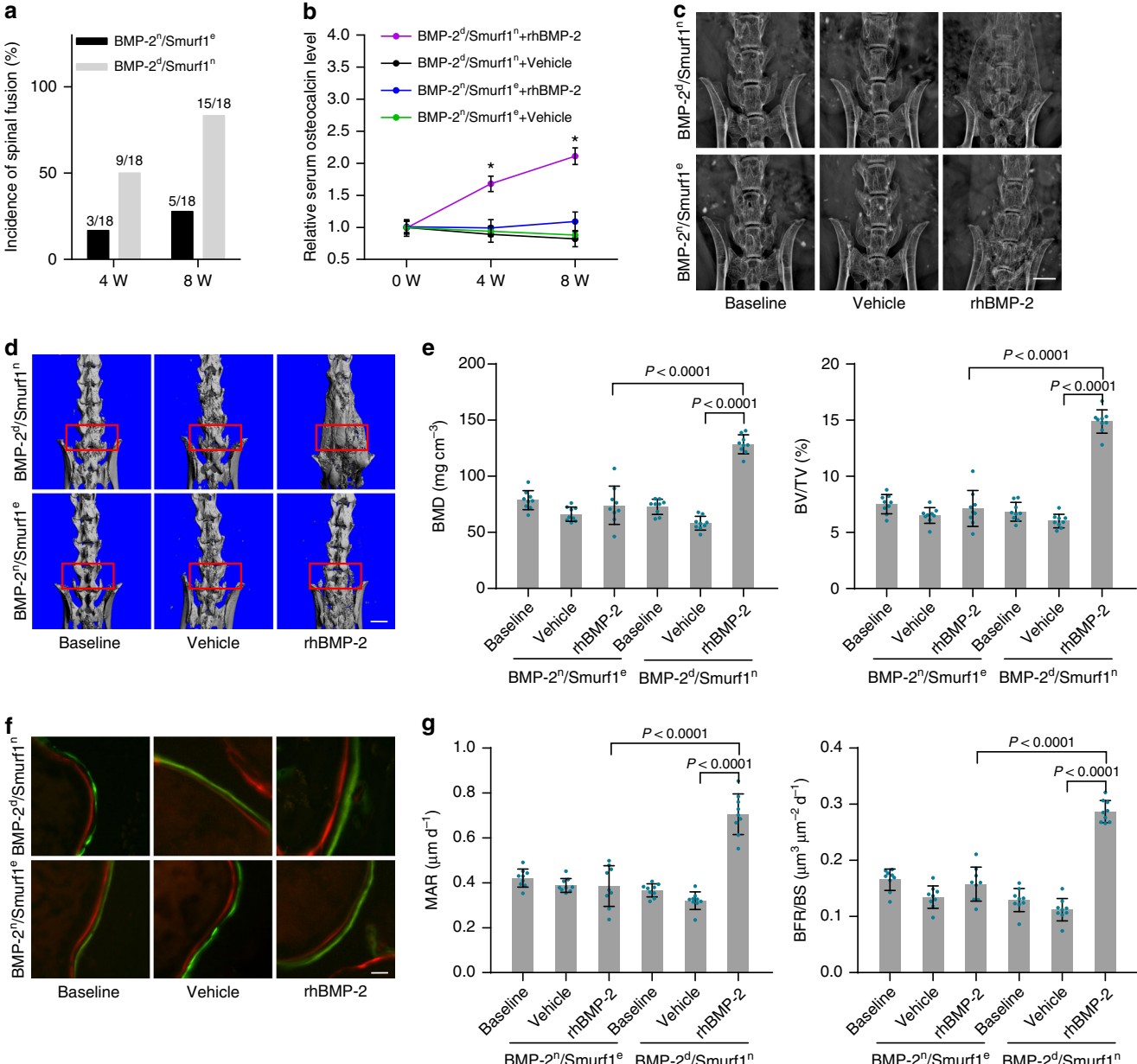

**Fig. 3** Local rhBMP-2 response in mouse BMP-2$^n$/Smurf1$^e$ and BMP-2$^d$/Smurf1$^n$ subgroups. **a** Manual assessment of spinal fusion incidence in BMP-2$^n$/Smurf1$^e$ and BMP-2$^d$/Smurf1$^n$ subgroups of 15-month-old osteoporotic mice (OVX at 6 months old) with local administration of rhBMP-2 (10 μg per piece of ACS bilaterally, $n = 18$ per group). Fisher's exact test was performed ($P = 0.002$ for BMP-2$^d$/Smurf1$^n$ versus BMP-2$^n$/Smurf1$^e$ at 8 W). **b** Relative serum osteocalcin level in BMP-2$^n$/Smurf1$^e$ and BMP-2$^d$/Smurf1$^n$ subgroups during local administration of rhBMP-2 or vehicle (PBS). The level of serum osteocalcin in BMP-2$^n$/Smurf1$^e$ and BMP-2$^d$/Smurf1$^n$ subgroups at 4 weeks (4 W) and 8 weeks (8 W) were normalized to their respective vehicle baseline at 0 week (0 W). *$P < 0.0001$ for BMP-2$^d$/Smurf1$^n$ + rhBMP-2 versus BMP-2$^n$/Smurf1$^e$ + rhBMP-2 at 4 W and 8 W. **c** Radiographic analysis of spinal fusion in L4-L6 vertebrae by X-ray. Scale bars, 3.0 mm. **d** Representative images showing intertransverse fusion bridge in spinal fusion sites by microCT reconstruction. Scale bars, 2.5 mm. **e** microCT measurements for BMD and BV/TV in spinal fusion sites (indicated by red rectangle). **f** Representative images showing bone formation in spinal fusion sites assessed by xylenol (red) and calcein (green) labeling. Scale bars, 10 μm. **g** Analysis of dynamic bone histomorphometric parameters (MAR and BFR/BS) in spinal fusion sites. $n = 9$ per group. Data are mean ± s.d. followed by one-way ANOVA with a post-hoc test

(Supplementary Fig. 6a). After in vitro incubation with rhBMP-2, no notable increase of p-Smad1, Runx2 activation and osteocalcin mRNA was found in osteoblasts from BMP-2$^n$/Smurf1$^e$ subgroup, while there was significant elevation of them in osteoblasts from BMP-2$^d$/Smurf1$^n$ subgroup (Supplementary Fig. 6b). Improvements of alkaline phosphatase (Alp) activity and mineralized nodule formation during osteogenic differentiation were not impressive in osteoblasts from BMP-2$^n$/Smurf1$^e$

subgroup when compared to those in osteoblasts from BMP-2$^d$/Smurf1$^n$ subgroup (Supplementary Fig. 6c and 6d). After Smurf1 gene silencing, we found decreased levels of Smurf1 activity (Smad1 bound to Smurf1, ubiquitination of Smad1 and Runx2) and enhanced levels of p-Smad1, Runx2 activation, osteocalcin mRNA, Alp activity, and mineralized nodule formation in osteoblasts from BMP-2$^d$/Smurf1$^n$ subgroup (Supplementary Fig. 6e and 6f).

**Virtual screening of inhibitors targeting Smurf1**. Smurf1 uses a coupled WW domain binding mechanism in its interaction with Smads[23,24]. The WW2 domain mainly responsible for the canonical binding with PY motif in Smads while the WW1 domain contacts the phosphorylation regions in linker of Smads, suggesting that WW1 and WW2 domains cooperate to maintain the functions of Smurf1[10,23–26]. The homology between human and mouse Smurf1 WW1-WW2 domains was 100% (Supplementary Fig. 7a). We performed homology modeling of Smurf1 WW1-WW2 domains (Fig. 4a, Supplementary Fig. 7b, c). Molecular docking was conducted between Smurf1 WW1-WW2 domains and Apollo Scientific Library. Considering binding affinity and drug-like criteria, we selected the 15 top-ranked small molecules as candidates (Supplementary Table 3). A chalcone derivative (2-(4-cinnamoylphenoxy)acetic acid) was chosen as the optimal small molecule duo to its maximal inhibition of Smurf1 activity (Smad1 bound to Smurf1, ubiquitination of Smad1, and Runx2) rather than Smurf1 expression, and enhancement of Runx2 activation, and osteocalcin mRNA level in osteoblasts from BMP-2$^n$/Smurf1$^e$ subgroup (Fig. 4b–d, Supplementary Fig. 8a, b). Inhibition of Smurf1 activity rather than Smurf1 expression, and enhancement of p-Smad1, Runx2 activation, osteocalcin mRNA, Alp activity, and mineralized nodule formation by the chalcone derivative were in dose-dependent manners (Fig. 4e–g, Supplementary Fig. 8c, d). No in vitro cytotoxicity of the chalcone derivative was observed in osteoblasts (Fig. 4h). The chalcone derivative had no effects on Smurf2 activity (Smad2 bound to Smurf2 and Smad2 ubiquitination)[27,28] (Supplementary Fig. 8e). Transforming growth factor-βs (TGF-βs) and BMPs belong to the same TGF-β superfamily[29,30]. TGF-βs activate canonical Smad2/3 signaling pathway, as well as non-canonical MAPK pathways (such as ERK1/2 and p38) to regulate numerous cellular processes[5,30,31]. The chalcone derivative did not affect the expression and activation of downstream molecules of TGF-βs including Smad2/3, p38, and ERK1/2 (Supplementary Fig. 8f), suggesting the high specificity of the chalcone derivative on Smurf1 inhibition and BMP signaling.

**Interaction between the chalcone derivative and Smurf1**. Six binding modes between the chalcone derivative and Smurf1 WW1-WW2 were predicted (Supplementary Fig. 7d). After incubation of osteoblasts from BMP-2$^n$/Smurf1$^e$ subgroup with vehicle (DMSO), the chalcone derivative or a negative control (resveratrol)[32], we performed immunoprecipitation using an anti-Smurf1 antibody and detected the existence of the chalcone derivative rather than resveratrol in immunoprecipitates (Supplementary Fig. 9a). We chose amino acid residues in the optimal mode including G248, R289, and Y297 for mutation studies (Supplementary Table 4). After immunoprecipitation using an anti-Flag antibody, we detected decreased level of the chalcone derivative in immunoprecipitates from osteoblasts overexpressing Flag-Smurf1 G248A, Y297A, or G248A/Y297A rather than R289A (Supplementary Fig. 9b). Level of the chalcone derivative decreased more obviously in immunoprecipitates from osteoblasts overexpressing Flag-Smurf1 G248A/Y297A (Supplementary Fig. 9b). We further conducted drug affinity responsive target stability (DARTS) assay, which relies on the protection of the target protein against proteolysis conferred by interaction with a small molecule[33,34]. It was performed by treating Flag-Smurf1, mutants, or BSA with the chalcone derivative or vehicle (DMSO), followed by digestion with protease subtilisin. Subsequently, the samples were separated on gel and stained to identify protein bands that were protected from proteolysis (Supplementary Fig. 9c). Mutation of G248 or Y297 significantly decrease the stability of Smurf1 conferred by the chalcone derivative. Mutation

of both G248 and Y297 lead to synergetic reduction of Smurf1 stability conferred by the chalcone derivative (Supplementary Fig. 9c). All the above results indicated that both G248 residue in WW1 domain and Y297 residue in WW2 domain were essential for chalcone derivative-Smurf1 interaction.

**Spinal fusion in BMP-2$^n$/Smurf1$^e$ mice**. BMP-2$^n$/Smurf1$^e$ subgroup of osteoporotic mice received posterolateral intertransverse lumbar fusion surgery in L4-L6 vertebrae and local administration of the chalcone derivative or vehicle (DMSO). For 15-month-old osteoporotic mice, there were significantly increased incidence of spinal fusion and serum osteocalcin after administration of the chalcone derivative when compared to vehicle (Fig. 5a, b). Radiographic observation, microCT, and bone histomorphometric analysis demonstrated that the chalcone derivative significantly enhanced fusion bridge, bone mass, and bone formation (Fig. 5c–g). Osteoblasts from L4-L6 vertebrae of mice administered with the chalcone derivative exhibited no obvious change of Smurf1 expression but inhibited Smurf1 activity (Smad1 bound to Smurf1, ubiquitination of Smad1 and Runx2) and increased p-Smad1, Runx2 activation, and osteocalcin mRNA levels (Supplementary Fig. 10a, b). We found no detectable in vivo toxicity of the chalcone derivative (Supplementary Table 5). Consistently, 8-month-old osteoporotic mice also showed increased spinal fusion incidence and serum osteocalcin, enhanced fusion bridge and bone mass, promoted bone formation, inhibited Smurf1 activity, and elevated p-Smad1, Runx2 activation, and osteocalcin mRNA levels after the local administration of the chalcone derivative (Supplementary Fig. 11).

**Association of the chalcone derivative with bone formation**. BMP-2$^n$/Smurf1$^e$ subgroup of osteoporotic mice received intravenous administration of the chalcone derivative at a series of doses (2.5, 5.0, 10.0, and 15.0 mg kg$^{-1}$) or vehicle (DMSO), with an injection interval of once every three days (Supplementary Fig. 12a). There was no significant improvement of bone formation parameters (MAR and BFR/BS) for trabecular bone in mice treated with thee chalcone derivative, when compared to those in vehicle-treated mice (Supplementary Fig. 12b). Further analysis showed no enhanced bone accumulation of the chalcone derivative with the increased doses (Supplementary Fig. 12c). However, a multiple linear regression analysis revealed a positive association of MAR with bone accumulation of the chalcone derivative rather than the injected doses (Supplementary Fig. 13).

**Conjugation of the chalcone derivative with (DSS)$_6$**. We conjugated the chalcone derivative with our previously identified osteoblast-targeting (DSS)$_6$[21] with a cell-penetrating property[22] (Supplementary Fig. 14a). Reverse phase high performance liquid chromatography (RP-HPLC) and mass spectrometry (MS) analysis showed a purity above 95% and reasonable molecular weight for (DSS)$_6$-chalcone derivative (Supplementary Fig. 14b, c). Osteoblasts from the BMP-2$^n$/Smurf1$^e$ subgroup were incubated with the chalcone derivative, (DSS)$_6$-chalcone derivative, (NAA)$_6$-chalcone derivative[35], or (DSS)$_6$ + chalcone derivative in the presence of rhBMP-2. (DSS)$_6$-chalcone derivative more notably inhibited Smurf1 activity (Smad1 bound to Smurf1, ubiquitination of Smad1 and Runx2), enhanced p-Smad1 level, Runx2 activation, and osteocalcin mRNA in osteoblasts when compared to other chalcone derivative formulations (Supplementary Fig. 15a, b). Cellular uptake of (DSS)$_6$-chalcone derivative was also higher than others (Supplementary Fig. 15c). Alp activity and mineralized nodule formation were more remarkably improved during differentiation of osteoblasts incubated with (DSS)$_6$-chalcone derivative (Supplementary Fig. 15d). There was

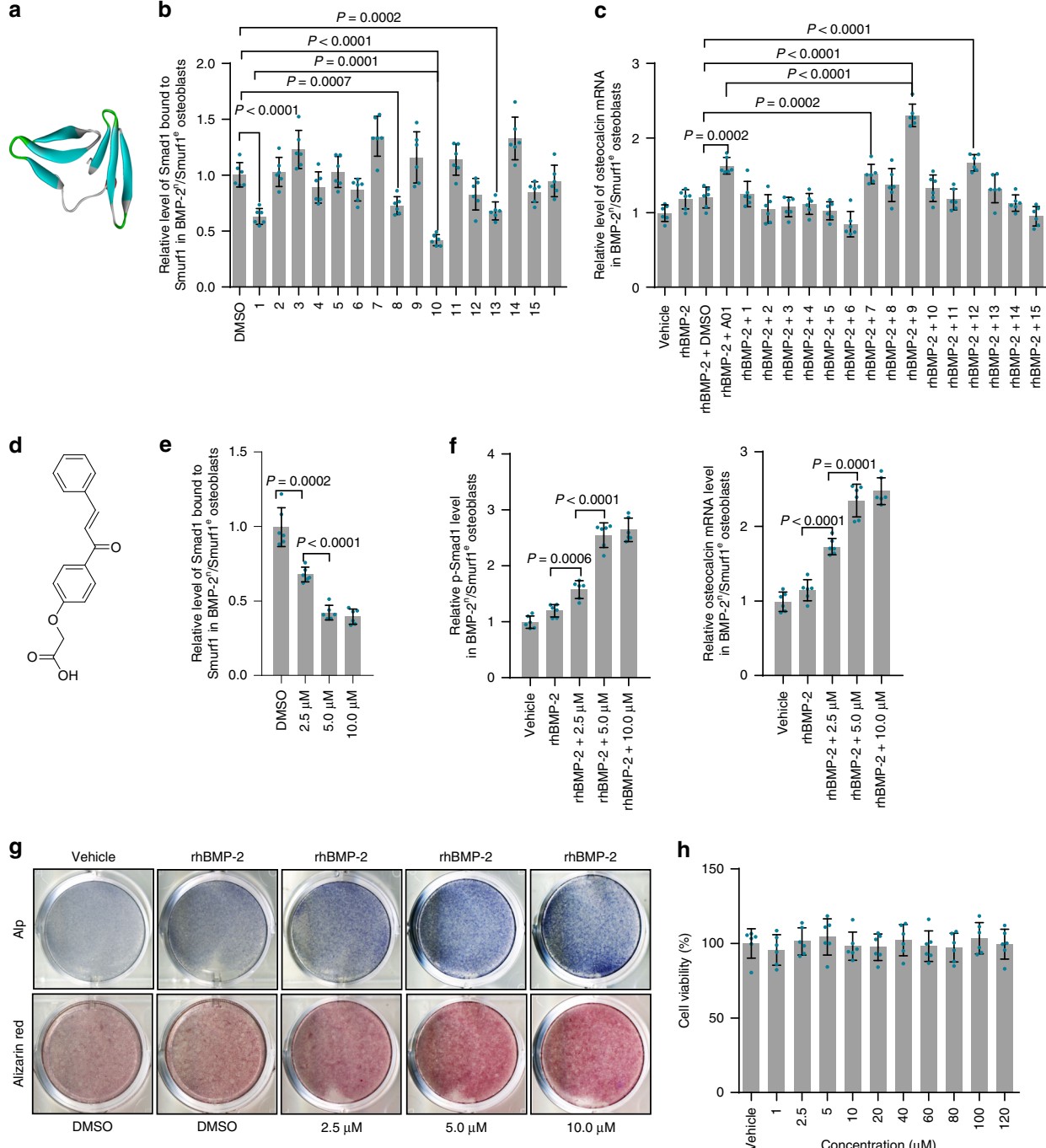

**Fig. 4** Virtual screening of Smurf1 small molecular inhibitors and in vitro effects of the inhibitors. **a** Homology modeling of Smurf1 WW1-WW2 domains. Left beta sheet, WW1 domain; right beta sheet, WW2 domain. **b** Effects of the top-ranked 15 small molecules (5.0 μM) on Smurf1 activity (Smad1 bound to Smurf1) in osteoblasts from BMP-2ⁿ/Smurf1ᵉ subgroup of 15-month-old osteoporotic mice (OVX at 6 months old). A01, a previously reported Smurf1 inhibitor. **c** Effects of the 15 small molecules (5.0 μM) on osteocalcin mRNA expression in the above osteoblasts with presence of 100 ng ml⁻¹ rhBMP-2 or vehicle (PBS). **d** Structural formula of the chalcone derivative (2-(4-cinnamoylphenoxy)acetic acid). **e** Relative level of Smad1 bound to Smurf1 in the above osteoblasts incubated with the chalcone derivative (2.5, 5.0, and 10.0 μM). **f** Relative levels of p-Smad1 and osteocalcin mRNA in the above osteoblasts incubated with the chalcone derivative with presence of 100 ng ml⁻¹ rhBMP-2 or vehicle (PBS). **g** Alp activity and mineralized nodule formation in the above osteoblasts incubated with the chalcone derivative with presence of 100 ng ml⁻¹ rhBMP-2 or vehicle (PBS). **h** In vitro cell viability of the chalcone derivative at a series of concentrations (1.0, 2.5, 5.0, 10.0, 20.0, 40.0, 60.0, 80.0, 100.0, and 120.0 μM). The levels of Smad1 bound to Smurf1, p-Smad1 and osteocalcin mRNA were normalized to the mean values of osteoblasts treated with vehicle (PBS) or DMSO. $n = 6$ per group. Data are mean ± s.d. followed by one-way ANOVA with a post-hoc test

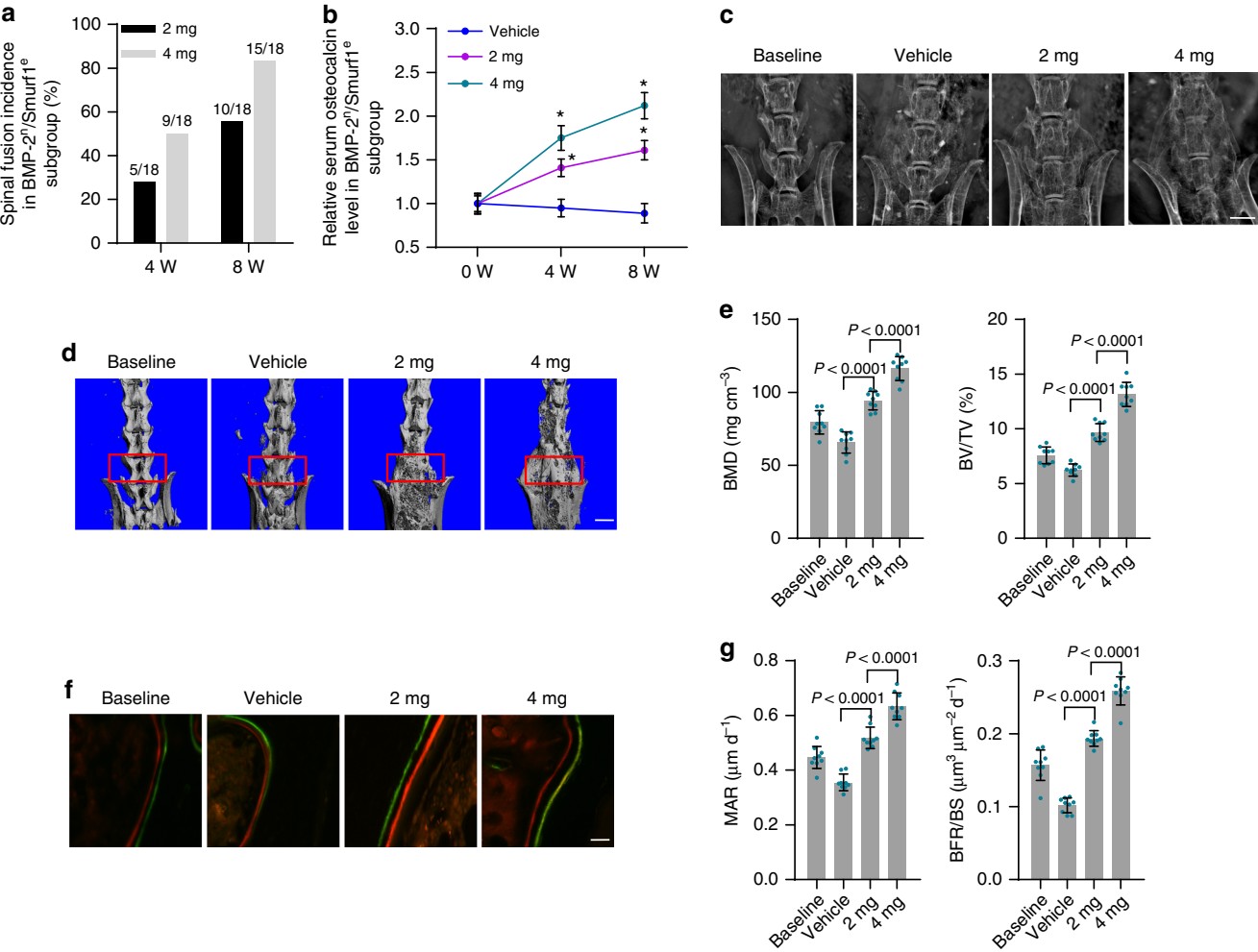

**Fig. 5** In vivo effects of the chalcone derivative on local bone formation during spinal fusion in BMP-2[n]/Smurf1[e] mice. **a** Manual assessment of spinal fusion incidence during local administration of the chalcone derivative (2 or 4 mg per piece of ACS bilaterally) in BMP-2[n]/Smurf1[e] subgroup of 15-month-old osteoporotic mice (OVX at 6 months old) at 4 weeks (4 W) or 8 weeks (8 W). $n = 18$ per group. Fisher's exact test was performed ($P = 0.016$ and $0.001$ for 2 mg and 4 mg versus vehicle at 4 W, respectively; $P < 0.0001$ for 2 mg or 4 mg versus vehicle at 8 W). **b** Relative serum osteocalcin level during local administration of the chalcone derivative or vehicle (DMSO) in BMP-2[n]/Smurf1[e] subgroup. The level of serum osteocalcin in BMP-2[n]/Smurf1[e] subgroup was normalized to the baseline before spinal fusion surgery. *$P < 0.0001$ for 2 mg versus vehicle or 4 mg versus 2 mg at 4 W and 8 W. **c** Radiographic analysis of spinal fusion in L4-L6 vertebrae by X-ray. Scale bars, 3.0 mm. **d** Representative images showing bone mass in spinal fusion sites (indicated by red rectangle) by microCT measurement. Scale bars, 2.5 mm. **e** microCT measurements for BMD and BV/TV in spinal fusion sites. **f** Representative images showing bone formation in spinal fusion sites assessed by xylenol (red) and calcein (green) labeling. Scale bars, 10 μm. **g** Analysis of dynamic bone histomorphometric parameters (MAR and BFR/BS) in spinal fusion sites. $n = 9$ per group. Data are mean ± s.d. followed by one-way ANOVA with a *post-hoc* test

no obvious in vitro cytotoxicity of (DSS)$_6$-chalcone derivative (Supplementary Fig. 15e).

**Bone/osteoblast accumulation and dose-response pattern.** BMP-2[n]/Smurf1[e] subgroup of osteoporotic mice received intravenous administration of chalcone derivative, (DSS)$_6$-chalcone derivative, (NAA)$_6$-chalcone derivative and (DSS)$_6$ + chalcone derivative, respectively. Distribution of chalcone derivative was significantly higher in bone but lower in liver and kidney in mice treated with (DSS)$_6$-chalcone derivative when compared to mice treated with other formulations (Supplementary Fig. 16a). After cell sorting, accumulation of (DSS)$_6$-chalcone derivative in osteoblasts (Alp$^+$ or osteocalcin$^+$ cells) was significantly higher than non-osteoblasts (Alp$^-$ or osteocalcin$^-$ or Oscar$^+$ cells) (Supplementary Fig. 16b). To determine dose-response pattern, BMP-2[n]/Smurf1[e] subgroup of osteoporotic mice received intravenous administration of the above formulations at doses ranking from 2.5 to 20.0 μmol

kg$^{-1}$. Smurf1 activity (Smad1 bound to Smurf1, ubiquitination of Smad1, and Runx2) decreased in a dose-dependent manner and reached the lowest level in osteoblasts from mice administered with (DSS)$_6$-chalcone derivative at a dose of 10.0 μmol kg$^{-1}$. However, mice treated with other chalcone derivative formulations showed no remarkable inhibition of Smurf1 activity (Supplementary Fig. 16c). After single injection of the above formulations at a dose of 10.0 μmol kg$^{-1}$, (DSS)$_6$-chalcone derivative exhibited the maximal inhibition of Smurf1 activity and enhancement of p-Smad1 level, Runx2 activation, and osteocalcin mRNA in osteoblasts at 24 h and maintained the effective action for 72 h, whereas other formulations showed no satisfactory effects (Supplementary Fig. 16d).

**Bone anabolic action in BMP-2[n]/Smurf1[e] mice.** BMP-2[n]/Smurf1[e] subgroup of osteoporotic mice received intravenous administration of the chalcone derivative formulations at a dose of 10.0 μmol kg$^{-1}$, with an injection interval of once every three

 

days, or daily subcutaneous injection of the clinically approved bone anabolic agent, i.e., recombinant human parathyroid hormone (PTH, amino acids 1–34) (teriparatide)[36–38] at a dose of 40.0 µg kg$^{-1}$ (Fig. 6a). For 15-month-old osteoporotic mice, better-organized microarchitecture, improved bone mass, and promoted bone formation for trabecular bone were observed after treatment with $(DSS)_6$-chalcone derivative or PTH, when compared to other chalcone derivative formulations (Fig. 6b–e). Bone

anabolic action of $(DSS)_6$-chalcone derivative was comparable to that of PTH (Fig. 6a–e). Osteoblasts from mice administered with $(DSS)_6$-chalcone derivative showed unchanged Smurf1 expression but inhibited Smurf1 activity (Smad1 bound to Smurf1, ubiquitination of Smad1, and Runx2) and increased p-Smad1, Runx2 activation, and osteocalcin mRNA levels when compared to those in osteoblasts from mice treated with other formulations (Supplementary Fig. 17a, b). Biochemistry and hematology assay

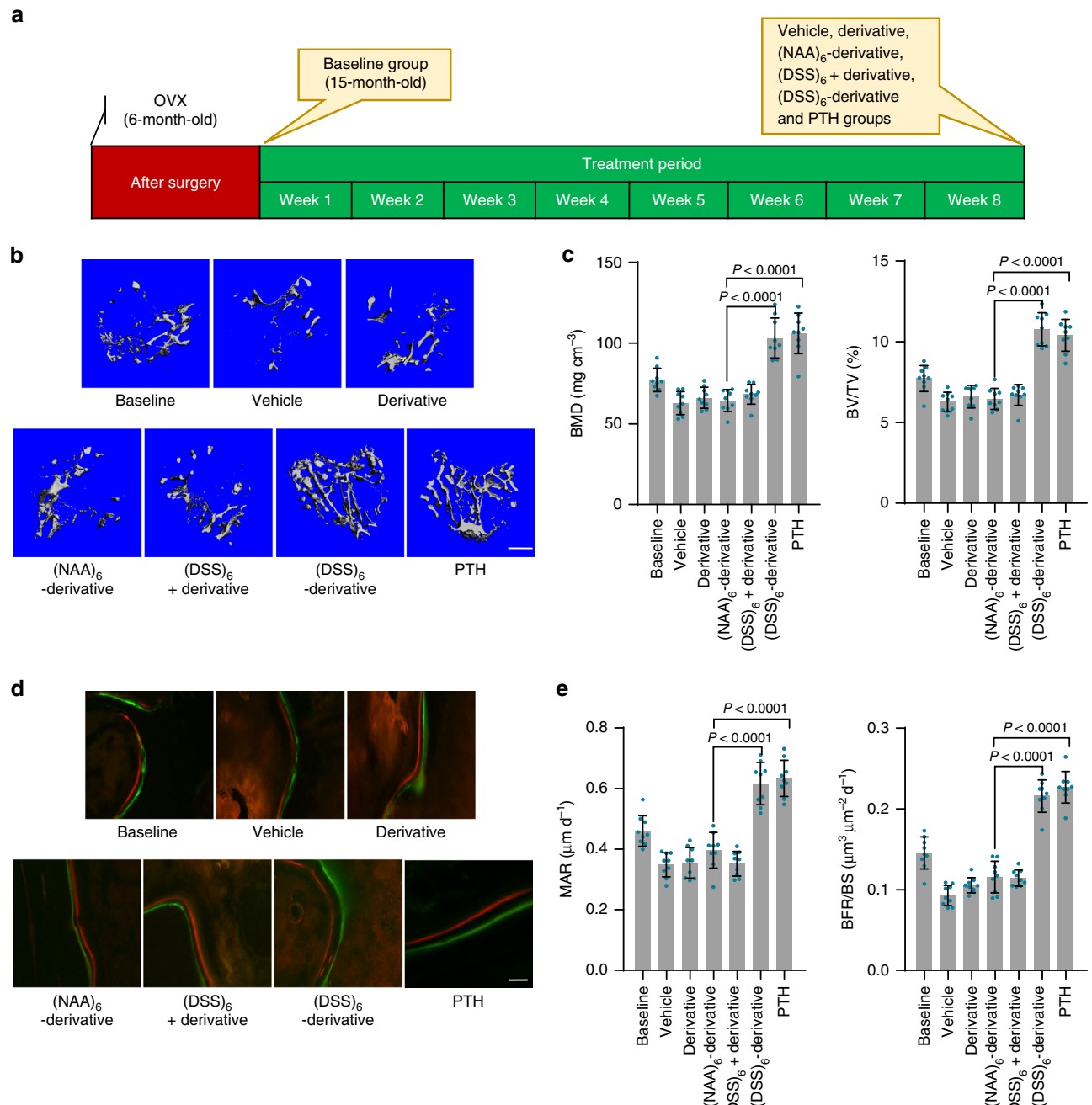

**Fig. 6** Systemic bone anabolic action of the chalcone derivative formulations in BMP-2$^n$/Smurf1$^e$ mice. **a** A schematic diagram illustrating the experimental design. 6-month-old mice were OVX and left untreated for 9 months (15-month-old) to induce osteoporosis. The mice received intravenous administration of different chalcone derivative formulations at a dose of 10.0 µmol kg$^{-1}$, with an injection interval of once every three days, or daily subcutaneous injection of recombinant human parathyroid hormone (PTH, amino acids 1–34) at a dose of 40.0 µg kg$^{-1}$. **b** Representative images showing three-dimensional trabecular architecture by microCT reconstruction at the proximal tibia. Scale bars, 500 µm. **c** microCT measurements for BMD and BV/TV at the proximal tibia. **d** Representative images showing bone formation at the proximal tibia assessed by xylenol (red) and calcein (green) labeling. Scale bar, 10 µm. **e** Analysis of dynamic bone histomorphometric parameters (MAR and BFR/BS) at the proximal tibia. $n = 9$ per group. Data are mean ± s.d. followed by one-way ANOVA with a post-hoc test

 

showed no detectable in vivo toxicity of the $(DSS)_6$-chalcone derivative (Supplementary Table 6). For 8-month-old BMP-2$^n$/Smurf1$^e$ subgroup of osteoporotic mice, $(DSS)_6$-chalcone derivative also showed comparable bone anabolic action with PTH, dramatically inhibited Smurf1 activity and increased p-Smad1, Runx2 activation, and osteocalcin mRNA in osteoblasts (Supplementary Fig. 18). However, no significant bone anabolic effects of $(DSS)_6$-chalcone derivative was observed in BMP-2$^d$/Smurf1$^n$ subgroup of osteoporotic mice (Supplementary Fig. 19).

## Discussion

In this study, we revealed that individuals (human and mice) with age-related osteoporosis could be classified into different subgroups based on distinct intraosseous levels of an extracellular BMP signaling initiator (BMP-2)[5] and ubiquitin ligase activity of an intracellular signaling suppressor (Smurf1)[7]. Both BMP-2$^d$/Smurf1$^n$ and BMP-2$^n$/Smurf1$^e$ subgroups had obviously reduced levels of intraosseous p-Smad1 and Runx2 activation and serum osteocalcin during aging, but the reduction extents in BMP-2$^d$/Smurf1$^n$ subgroup was more significant. This might be explained by the different regulatory modes of BMPs and Smurf1 on the signaling transduction[7], in which BMPs govern initiation of signaling, while Smurf1 inhibits downstream molecules by a negative feedback loop[13,39].

Regarding the utility of serum osteocalcin in clinical evaluation of osteoporosis[40] and discriminated reduction extents of serum osteocalcin between BMP-2$^d$/Smurf1$^n$ and BMP-2$^n$/Smurf1$^e$ subgroups, we speculated that serum osteocalcin could be a biomarker in stratifying the two subgroups before bone anabolic therapy. As expected, we classified two major subgroups according to the serum osteocalcin-based classification methodology, in which one subgroup showed a decreased BMP-2 level and normal Smurf1 activity as those in BMP-2$^d$/Smurf1$^n$ subgroup, while another subgroup demonstrated a normal BMP-2 level and elevated Smurf1 activity as those in BMP-2$^n$/Smurf1$^e$ subgroup.

After local administration with rhBMP-2, we found enhanced bone formation during spinal fusion in BMP-2$^d$/Smurf1$^n$ subgroup but poor bone anabolic response in BMP-2$^n$/Smurf1$^e$ subgroup. These results not only provided a possible explanation for the large inter-individual variations in therapeutic efficacy of rhBMPs, but also implied that supplement of rhBMP-2 was not an efficient approach to promote bone formation in BMP-2$^n$/Smurf1$^e$ subgroup. By genetic down-modulation of Smurf1, our in vitro data demonstrated enhanced osteoblastic BMP signaling and osteogenic differentiation in BMP-2$^n$/Smurf1$^e$ subgroup, indicating that inhibition of Smurf1 activity could be an alternative bone anabolic approach in BMP-2$^n$/Smurf1$^e$ subgroup.

To achieve pharmacological inhibition of Smurf1 activity, we performed virtual screening to select small molecular inhibitors targeting both Smurf1 WW1 and WW2 domains. After the molecular docking, a chalcone derivative, i.e., 2-(4-cinnamoyl-phenoxy)acetic acid, showed maximal effects on inhibition of Smurf1 activity when compared to other candidates and previously reported small molecular inhibitor A01, which was screened to target only Smurf1 WW1 domain[8]. Mutation studies and DARTS assay confirmed that the chalcone derivative could interact with both Smurf1 WW1 and WW2 domains. Our in vitro and in vivo data consistently demonstrated that the chalcone derivative remarkably increased BMP signaling and promoted osteogenic differentiation, leading to improved local bone formation during spinal fusion in BMP-2$^n$/Smurf1$^e$ subgroup of osteoporotic mice with no detectable toxicity.

After systemic administration of the chalcone derivative, we observed no significantly enhanced bone formation in BMP-2$^n$/Smurf1$^e$ subgroup of osteoporotic mice. However, further analysis showed a positive association between bone formation and bone distribution of the chalcone derivative, suggesting that the unsatisfactory bone anabolic action was due to the inadequate bone distribution of the chalcone derivative. Thus, we conjugated the chalcone derivative to an osteoblast-targeting and cell-penetrating oligopeptide $(DSS)_6$[21,22]. $(DSS)_6$ facilitated the conjugated chalcone derivative entering osteoblasts, decreasing Smurf1 activity, increasing BMP signaling and promoting osteogenic differentiation in osteoblasts from BMP-2$^n$/Smurf1$^e$ subgroup in vitro. In vivo data demonstrated that $(DSS)_6$ facilitated the conjugated chalcone derivative targeting osteoblasts, leading to promoted systemic bone formation in BMP-2$^n$/Smurf1$^e$ subgroup. Notably, bone anabolic action of $(DSS)_6$-chalcone derivative in BMP-2$^n$/Smurf1$^e$ subgroup was comparable to that of the clinically approved PTH[36–38]. However, we didn't observe the systemic bone anabolic action of the $(DSS)_6$-chalcone derivative in BMP-2$^d$/Smurf1$^n$ subgroup, implying that the satisfactory bone anabolic action of the $(DSS)_6$-chalcone derivative was specific for BMP-2$^n$/Smurf1$^e$ subgroup.

All the above results indicate that inhibition of osteoblastic Smurf1 could be a precision medicine-based bone anabolic strategy for BMP-2$^n$/Smurf1$^e$ subgroup of age-related osteoporotic individuals. The chalcone derivative targeting Smurf1 demonstrates great potential to enhance local bone formation during spinal fusion in BMP-2$^n$/Smurf1$^e$ subgroup. $(DSS)_6$-chalcone derivative conjugate could be an alternative systemic bone anabolic agent for BMP-2$^n$/Smurf1$^e$ subgroup (Supplementary Fig. 20).

## Methods

**Collection of human bone specimens and blood samples**. We collected human bone specimens and blood samples from the Shenzhen People's Hospital. Aged osteoporotic VCF patients ($n = 75$ for 60–69 years old; $n = 75$ for 70–79 years old), adult traumatic VCF patients ($n = 41$ for 30–39 years old), aged osteoporotic LSS patients ($n = 74$ for 60–69 years old) and adult LDH patients ($n = 39$ for 30–39 years old) were included in our study (inclusive criteria). Subjects with malignancy, diabetes or other severe diseases in the previous five years were excluded from our study (exclusive criteria). The collected bone specimens and blood samples were preserved in liquid nitrogen and the liquid nitrogen was periodically replenished. Before the biochemical examination, the non-bone tissues (soft tissues and blood) and cortical bone were removed from the bone specimens. The assays were performed by one well-trained researcher in all the laboratory batches. The sample collection and handling procedures were identical for all patients. For quality control, the bone specimens were detected for positive expression of bone turnover markers including collagen I (Col-1, a bone formation marker)[41] and tartrate-resistant acid phosphatase (TRAP, a bone resorption marker)[42] by real-time PCR. All the clinical procedures were approved by the Committees of Clinical Ethics in the Shenzhen People's Hospital. We obtained informed consent from the participants.

**Osteoporotic mouse model**. All the female C57BL/6J mice were maintained under standard animal housing conditions (12-h light, 12-h dark cycles and free access to food and water). The mice were ovariectomized (OVX) at 6 months old and left untreated for 2 months (8-month-old), 9 months (15-month-old) or 12 months (18-month-old). Osteoporosis in OVX mice were confirmed by microCT analysis before further investigation. At the corresponding time points in each study, we collected bone specimens (bilateral femurs/tibias and vertebra) and blood samples from osteoporotic mice and healthy adult 6-month-old mice. All the experimental procedures were approved by the Committees of Animal Ethics and Experimental Safety of Hong Kong Baptist University[43,44].

**Posterolateral intertransverse lumbar fusion mouse model**. Osteoporotic mice were anaesthetized and received posterolateral lumbar spine fusion surgery[45,46]. Briefly, Hair overlying the operative site of mice was shaved and the area was prepped with iodine. Mice were positioned prone with folded gauze beneath the abdomen. With the intention of fusing the lumbar spine in L4-L6 vertebrae, surface anatomy was located to determine incision point. The iliac crest is approximately level with the L5-L6 interspace. The procedure was performed under an operating microscope at 2.5 × magnification. A 15 mm incision was made in the skin along the midline, centered over a line running between the iliac crests. The skin was retracted and held with a self-retaining retractor. The paravertebral muscles overlying the articular processes of L4-L6 were separated from the spinal column

by scraping a 10 mm blade down the lateral border of the spinous process and pulling the muscles laterally. A pneumatic 1 mm round tip diamond burr was used to decorticate the visible articular processes until punctate bleeding was observed. Two pieces of absorbable collagen sponge (ACS; 2 mm × 7 mm) containing rhBMP-2 (10 μg per piece of ACS) or the chalcone derivative (2 or 4 mg per piece of ACS) or vehicle (PBS or DMSO) were placed bilaterally adjacent to the decorticated bone. The fascia was closed with a single line of continuous sutures using 6.0 Vicryl and the skin was subsequently closed in the same fashion. Mice were placed on a heat pad following surgery and monitored for recovery. Antibiotics were administered in the drinking water for the duration of the experiment.

**Isolation of primary osteoblasts**. Osteoblasts were isolated from bone pieces of mouse femurs or L4-L6 vertebrae by enzymatic digestion with α-MEM medium (Life Technologies) containing 0.1% collagenase (Life Technologies) and 0.25% trypsin (Life Technologies)[47]. Osteoblastic phenotype was validated by microscopical appearance and high mRNA expression of osteogenic markers (Alp, Col 1 and Runx2) determined by real-time PCR[41]. Absence of staining for von Willebrand factor (factor VIII) showed no notable contamination of endothelial cells[47]. No mycoplasma contamination was observed.

**Fluorescencge-activated cell sorting (FACS)**. Bone marrow cells were harvested from mouse femurs and tibias. Goat polyclonal Alp primary antibody (2.5 μg per $10^6$ cells, AF2910, R&D Systems), rabbit polyclonal osteocalcin antibody (1: 10, FL-100, Santa Cruz Biotechnology) and rabbit polyclonal Oscar antibody (1: 10, H-94, Santa Cruz Biotechnology) were used for FACS. Briefly, after washing with PBS containing 1% BSA, the bone marrow cells were incubated with the antibody to Alp or osteocalcin or Oscar before being stained with donkey anti-goat FITC-IgG (1: 200, ab6881, Abcam) or donkey anti-rabbit FITC-IgG (1: 100, sc-2090, Santa Cruz Biotechnology)[43,44,48]. The obtained cell populations were washed three times and used for quantitative analysis of chalcone derivative formulations and determination of Smurf1 activity, p-Smad1 and osteocalcin mRNA.

**Enzyme-linked immunosorbent assay (ELISA)**. Human and mouse BMP-2 in bone specimens was determined by BMP-2 ELISA kits from R&D Systems and Abcam, respectively, according to the manufacturer's instructions. Smurf1 in human and mouse bone specimens was examined using anti-Smurf1 antibody (1: 1000, ab38866, Abcam) following established ELISA protocol (TECH TIP #65, Thermo Scientific). p-Smad1 in human and mouse bone specimens or mouse osteoblasts was examined by a p-Smad1 ELISA kit from Abcam according to the manufacturer's instruction. Human and mouse serum osteocalcin was quantified by osteocalcin ELISA kits from R&D Systems and ALPCO, respectively, according to the manufacturer's instructions. Lyophilized BMP-2, Smurf1, p-Smad1 and osteocalcin were used as controls in ELISA, respectively. Runx2 activation in human and mouse bone specimens was determined by a Runx2 transcription factor assay kit from Abcam, according to the manufacturer's instruction.

**Immunoprecipitation in combination with ELISA**. Human and mouse bone extracts or mouse osteoblast lysates were prepared in HEPES lysis buffer (20 mM HEPES pH 7.2, 50 mM NaCl, 0.5% Triton X-100, 1 mM NaF, 1 mM dithiothreitol) supplemented with protease inhibitor cocktail (Roche Life Science) and phosphatase inhibitors (10 mM NaF and 1 mM $Na_3VO_4$). Immunoprecipitation was performed using an anti-Smurf1 antibody (1: 50, 45-K, Santa Cruz Biotechnology) or an anti-Smurf2 antibody (1: 50, D-5, Santa Cruz Biotechnology) and protein A/G-agarose (Santa Cruz Biotechnology) at 4 °C[49]. For detection of Smad1 bound to Smurf1, Smad1 level in immunoprecipitates was quantified by a Smad1 ELISA kit from Abcam according to the manufacturer's instruction. For detection of Smad2 bound to Smurf2, Smad2 level in immunoprecipitates was quantified by a Smad2 ELISA kit from Aviva Systems Biology according to the manufacturer's instruction.

**Ubiquitination assay in combination with ELISA**. Human bone extracts were prepared in HEPES lysis buffer (20 mM HEPES pH 7.2, 50 mM NaCl, 0.5% Triton X-100, 1 mM NaF, 1 mM dithiothreitol) supplemented with protease inhibitor cocktail (Roche Life Science), phosphatase inhibitors (10 mM NaF and 1 mM $Na_3VO_4$) and proteasome inhibitor (MG132). Smad1, Smad2 and Runx2 were immunoprecipitated with anti-Smad1 antibody (1: 100, PA5-17122, Invitrogen), anti-Smad2 (1: 50, 5339, Cell Signaling Technology) and anti-Runx2 antibody (1: 50, 12556, Cell Signaling Technology), respectively, and protein A/G-agarose (Santa Cruz Biotechnology) at 4 °C. Ubiquitination of Smad1, Smad2 and Runx2 in immunoprecipitates was determined by a ubiquitin ELISA kit from Sigma according to the manufacturer's instruction.

**Molecular docking-based virtual screening**. Position-Specific Iterated BLAST (PSI-BLAST) against PDB database was performed to identify homology templates of Smurf1 WW1-WW2 domains. Homology modeling of Smurf1 WW1-WW2 domains were performed by MODELER software[50] based on structures of Smurf2 (PDB code: 2KXQ [https://www.rcsb.org/structure/2KXQ], identity: 86%), Smurf1 WW1 domain (PDB code: 2LAZ [https://www.rcsb.org/structure/2laz]) and Smurf1 WW2 domain (PDB code: 2LB1 [https://www.rcsb.org/structure/2lb1]).

Structure with lowest energy was chosen as the starting point for further structure refinements. In order to find optimized conformations, loop refinement was applied to the predicted structure of Smurf1 WW1-WW2 domains[51]. Ramachandran plot was used to estimate the quality of modeled structure by inspecting backbone dihedral angles ψ against φ of amino acid residues in protein structure[52]. Molecular docking-based virtual screening was conducted between Smurf1 WW1-WW2 domains and Apollo Scientific Library containing 43007 small molecules. Virtual screening parameters were prepared by autodock tools (ADT). The whole surface on Smurf1 WW1-WW2 domains were used as binding sites to screen small molecules with low binding energy.

**Cell culture and reagents**. The isolated osteoblasts were cultured in α-MEM medium with 10% fetal bovine serum (FBS) (Life Technologies) and 1% penicillin-streptomycin (Life Technologies). The cells were maintained under standard cell culture conditions of 5% $CO_2$ and 95% humidity. For in vitro experiments, confluent cells were removed using 0.25% trypsin containing 10 mM EDTA, centrifuged at 200 × g, resuspended in antibiotic-free growth medium and plated onto six-well plates or flasks. The chalcone derivative (2-(4-cinnamoylphenoxy)acetic acid) with a purity above 98% was synthesized in HitGen LTD.. (DSS)₆-chalcone derivative, (NAA)₆-chalcone derivative and oligopeptide (DSS)₆ with a purity above 95% were synthesized in ChinaPeptides Corporation. Human PTH (1–34) was purchased from Bachem, Torrance, CA.

**In vitro osteoblastic differentiation and treatment**. To examine osteogenic differentiation induced by rhBMP-2, mouse osteoblasts were cultured in α-MEM medium containing rhBMP-2 (100 ng ml⁻¹) (R&D Systems) or vehicle (PBS). The medium was changed every three days. 48 h after the incubation, the cells were harvested for determining levels of p-Smad1, Runx2 activation and osteocalcin mRNA. 6 days after the incubation, Alp staining was performed using an Alp detection kit (Sigma) according to the manufacturer's protocol. 13 days after the incubation, cells were treated with alizarin red staining for visualizing mineralized nodule formation. To determine effects of Smurf1 silencing on osteogenic differentiation, mouse osteoblasts were transfected with Smurf1 siRNA or non-sense siRNA (NC siRNA) and simultaneously administrated with rhBMP-2 (100 ng ml⁻¹), with an interval of once every three days. 48 h after the incubation, levels of Smurf1, Smurf1 activity (Smad1 bound to Smurf1 and ubiquitination of Smad1 and Runx2), Runx2 activation, p-Smad1, osteocalcin mRNA were determined. 6 days and 13 days after the incubation, Alp activity and formation of mineralized nodules were examined, respectively. To determine effects of different chalcone derivative formulations or other small molecules on osteogenic differentiation, mouse osteoblasts were treated with chalcone derivative formulations or other small molecules in presence of rhBMP-2 (100 ng ml⁻¹), with an interval of once every three days. 48 h after the treatment, levels of Smurf1, Smurf1 activity, p-Smad1, Runx2 activation, osteocalcin mRNA were determined. 6 days and 13 days after the incubation, Alp activity and formation of mineralized nodules were examined, respectively.

**Construction of expression vectors**. A cDNA construct containing the full-length open reading frame of Smurf1 wild type was subcloned into Flag-tagged mammalian expression vector pcDNA3.1 (Addgene). Site mutations including R289A, G248A, Y297A and G248A/Y297A in Flag-Smurf1 were made by the QuikChange site-directed mutagenesis kit (Agilent Technologies) and subcloned into Flag-tagged mammalian expression vector pcDNA3.1 (Addgene). Both Smurf1 wild type and mutants were verified by DNA sequencing analysis. The expression vectors were transfected into mouse osteoblasts using X-tremeGENE™ HP DNA Transfection Reagent (Roche Life Science), according to the manufacturer's instructions. Flag-tagged Smurf1 wild type and mutants were purified by a FLAG® M purification kit (Sigma), according to the manufacturer's instruction.

**Immunoprecipitation**. Mouse osteoblasts were incubated with the chalcone derivative (20 μM) or resveratrol (20 μM) or vehicle (DMSO) for 12 h. Cell lysates were prepared in HEPES lysis buffer (20 mM HEPES pH 7.2, 50 mM NaCl, 0.5% Triton X-100, 1 mM NaF, 1 mM dithiothreitol) supplemented with protease inhibitor cocktail (Roche Life Science) and phosphatase inhibitors (10 mM NaF and 1 mM $Na_3VO_4$). Immunoprecipitation was performed using an anti-Smurf1 antibody (1: 50, 45-K, Santa Cruz Biotechnology) and protein A/G-agarose (Santa Cruz Biotechnology) at 4 °C[25,49]. For mutation studies, mouse osteoblasts were transfected with Flag-Smurf1 or Flag-Smurf1 mutants for 40 h and then incubated with the chalcone derivative (20 μM) or vehicle (DMSO) for 12 h. Cell lysates were prepared and immunoprecipitation was performed using an anti-Flag antibody (5.0 mg/ml, F7425, Sigma) and protein A/G-agarose (Santa Cruz Biotechnology) at 4 °C[25,49].

**Western blots**. Mouse osteoblasts were incubated with the chalcone derivative at different concentrations (0, 2.5, 5.0 and 10.0 μM). 48 h after the incubation, total proteins were extracted from the cells and quantified using Bradford Assay (Bio-Rad). Protein samples were separated by sodium dodecyl sulfate-polyacrylamide gel electrophoresis and transferred onto PVDF membranes (Bio-Rad). After blocking, the membranes were probed with primary antibodies and then incubated with specific horseradish peroxidase-conjugated secondary antibodies (Bio-Rad).

Immunodetection was performed using enhanced chemiluminescence consistent with the manufacturer's protocol (Thermo Fisher Scientific). Primary antibodies including anti-ERK1/2 (1: 1000, 137F5), anti-p-ERK1/2 (1: 1000, 9101), anti-Smad2 (1: 1000, 5339), anti-p-Smad2 (1: 1000, E8F3R), anti-Smad3 (1: 1000, 9523), anti-p-Smad3 (1: 1000, E8F3R), anti-p38 (1: 1000, 8690), anti-p-p38 (1: 1000, 4511) and anti-β-actin (1: 1000, 3700) were purchased from Cell Signaling Technology. The band intensities were quantified and normalized to the corresponding controls. β-actin was used as a loading control for internal correction[53]. The uncropped blots were shown in Supplementary Fig. 21.

**Liquid chromatography-tandem mass spectrometry (LC-MS/MS).** To determine in vitro cellular uptake, interaction of small molecules with Smurf1, bone distribution and osteoblast accumulation chalcone derivative formulations, small molecules in immunoprecipitates or chalcone derivative formulations in osteoblasts or bone extraction were extracted by 90% methanol (Sigma) and analyzed by Agilent 6400 Ultra High Performance Liquid Chromatograph with Tripe Qquadrupole Mass Spectrometer (UHPLC-QqQ MS/MS) (Agilent Technologies)[54].

**Drug affinity responsive target stability (DARTS) assay.** Flag-Smurf1 wild type, mutants or BSA in 100 μl PBS was incubated with DMSO, the chalcone derivative (5.0 and 10.0 μM) for 18 h at 4 °C and then digested with subtilisin at room temperature (Sigma) for 20 min. The reactions were stopped by adding SDS loading buffer and boiling for 5 min. Samples were loaded onto a 12% acrylamide SDS-PAGE gel and then stained with coomassie brilliant blue to visualize the banding pattern[33,34]. The uncropped gel images were shown in Supplementary Fig. 21.

**Real-time PCR.** A RNeasy Mini Kit (Qiagen) was used to extract total RNA using the manufacturer's protocol. Total RNA was reverse transcribed into cDNA using a QuantiTect Reverse Transcription Kit according to an established protocol (Qiagen). The 10 μl volume of the final real-time PCR solution contained 1 μl diluted cDNA product, 5 μl 2 × Power SYBR® Green PCR Master Mix (Applied Biosystems), 0.5 μl each of forward and reverse primers and 3 μl nuclease-free water. The forward and reverse primers[41] were used for determining Alp, Col-1, Runx2 and osteocalcin. Mouse Alp: 5′-ATCTTTGGTCTGGCTCCCATG-3′ (forward), 5′-TTTCCCGTTCACCGTCCAC-3′ (reverse). Mouse Col-1: 5′-CCTGGTAAAGAT GGTGCC-3′ (forward), 5′-CACCAGGTTCACCTTTCGCACC-3′ (reverse). Mouse Runx2: 5′-GCATGGTGGAGGTACTAGCTG-3′ (forward), 5′-GCCGTCCACTGT GATTTTG-3′ (reverse). Mouse osteocalcin: 5′-GCAATAAGGTAGTGAACAGAC TCC-3′ (forward), 5′-GTTTGTAGGCGGTCTTCAAGC-3′ (reverse). Mouse GAPDH: 5′-TGCACCACCAACTGCTTAG-3′ (forward), 5′-GGATGCAGGGAT GATGTTC-3′ (reverse). The fluorescence signal emitted was collected by an ABI PRISM® 7900HT Sequence Detection System and the signal was converted into numerical values by SDS 2.1 software (Applied Biosystems)[21,48].

**Alkaline phosphatase staining.** Alkaline phosphatase staining was monitored using a fast violet B salt kit (procedure number 85, Sigma Aldrich)[55]. Briefly, one fast violet B salt capsule was dissolved in 48 ml of distilled water and 2 ml of naphtol AS-MX phosphate alkaline solution. Cells were fixed by immersion in a citrate-buffered acetone solution (two parts citrate and three parts acetone) for 30 s and rinsed in deionized water for 45 s. The samples were then placed in an alkaline phosphatase stain for 30 min. The whole procedure was protected from light. After 2 min of rinsing in deionized water, slides were treated with Mayer's hematoxylin solution for 10 min[43].

**Alizarin red staining.** Osteoblasts were fixed in 70% ice-cold ethanol for 1 h and rinsed with double-distilled H₂O. Cells were stained with 40 mM Alizarin Red S (Sigma), pH 4.0, for 15 min with gentle agitation. Cells were rinsed five times with double-distilled H₂O and then rinsed for 15 min with 1 × PBS while gently agitating[43].

**Manual assessment of spinal fusion.** At 4 or 8 weeks after spinal fusion surgery, mice were euthanized, and the spines were surgically removed. Three independent observers assessed the spines for movement within the L4 and L6 intervertebral space by manual palpation and visualization. Nonunion was recorded if motion was observed between the facets or transverse processes on either side. Complete fusion was recorded if no motion was observed bilaterally. The observers were blinded to the group allocation. Spines were scored as either fused or not fused. The L4-L6 segments were considered to be fused only when all three observers agreed[56–58].

**Radiographic assessment of spinal fusion.** Spinal fusion in L4-L6 vertebrae was evaluated with radiographs by an MX-20 Specimen Radiography System (Faxitron X-Ray) under consistent conditions (35 kV, 300 mA, 300 s). Fusion was considered to have occurred when there was clear evidence of new bone formation and osseous bridging with cortical continuity between the L4 and L6 transverse processes[56–59].

**microCT analysis.** For the proximal tibias, the whole secondary spongiosa at the left proximal tibias from each mouse was scanned ex vivo using a microCT system (vivaCT40, SCANCO MEDICAL). Briefly, 200 slices with a voxel size of 10 μm were scanned at the region of the proximal tibias beginning at the growth plate and extending distally along the tibia diaphysis. Eighty continuous slices beginning at 0.1 mm from the most distally aspect of the growth plate in which both condyles were no longer visible were selected for analysis. For the lumbar vertebral bodies in spinal fusion sites, the entire region of secondary spongiosa between the proximal and distal aspects from L5 vertebra was scanned by microCT with a voxel size of 10 μm. 350 slices, which included the entire region of the L5 body, were analyzed. All trabecular bone from each selected slice was segmented for three-dimensional reconstruction (sigma = 1.2, supports = 2 and threshold = 200) to calculate the bone formation parameters including BMD and BV/TV[43,44].

**Bone histomorphometric analysis.** The mice were injected intraperitoneally with xylenol orange (30 mg kg⁻¹) and calcein green (10 mg kg⁻¹) at 10 and 2 days, respectively, before euthanasia. After a microCT scan, the proximal tibia and L5 vertebra were dehydrated in graded concentrations of ethanol and embedded without decalcification in modified methyl methacrylate[21]. Frontal sections of trabecular bone at a thickness of 15 μm was obtained from the proximal tibia or L5 vertebra with an EXAKT Cut/grinding System (EXAKT Technologies). Dynamic histomorphometric analyses for bone formation parameters including MAR and BFR/BS) were performed using professional image analysis software (BIOQUANT OSTEO analysis software, Version 13.2.6) under fluorescence microscope (Leica image analysis system, Q500MC). The bone parameters were calculated and expressed according to standardized nomenclature for bone histomorphometry[48].

**Assay for biochemistry and hematology parameters.** Blood samples were collected from mice after treatment with the chalcone derivative or (DSS)₆-chalcone derivative or vehicle (DMSO). Serum biochemistry parameters including alanine transaminase (ALT), aspartate transaminase (AST) and blood urea nitrogen (BUN) were analyzed by a Vitros 250 Analyzer (Ortho Clinical Diagnostics, Johnson & Johnson Co, Rochester, NY). Hematology parameters including red blood cell (RBC), hemoglobin (HGB), hematocrit (HCT), white blood cell (WBC) and platelets (PLT) were determined by the fully automated ABX Pentra 60 C + Analyzer (Horiba ABX, Montpellier, France)[48].

**Statistical and data analysis.** Data are mean ± standard deviation (s.d.) followed by one-way analysis of variance (ANOVA) with a *post-hoc* test (for difference among multiple independent groups) or Student's *t* test (for difference between two independent groups). Categorical data were analyzed by chi-squared test or Fisher's exact test, if appropriate. All the statistical analyses were performed with a SPSS software, version 22.0. $P < 0.05$ was considered statistically significant. All the statistical data were analyzed by a contract service from Bioinformedicine (San Diego, CA, USA). Primary osteoblasts were used for the in vitro studies, such as ALP activity, mineralization nodule formation and Smurf1 activity assay. To avoid introducing bias, we conducted preliminary studies to determine appropriate cell concentrations. Two types of replicates including biological replicates (replicates of mice) and technical replicates (replicates within one mouse) was used, to yield more accurate and reliable statistics. Biological replicates were parallel measurements of biologically distinct samples that captured random biological variation. For example, each in vitro study was performed six times using osteoblasts isolated from six distinct mice. Technical replicates were repeated measurements of the same sample that represented independent measures of the random noise associated with protocols or equipment (i.e., three within-mouse replicates). For in vivo experiments, the sample size was calculated by the formula: $n = 2 [(U_\alpha + U_\beta) \, S \, \delta^{-1}]^2$. $S$ in the formula referred to the s.d. $\delta$ in the formula represented the inter-group difference of the mean values. There was a difference above 30 mg cm⁻³ in BMD between (DSS)₆-chalcone derivative group and (NAA)₆-chalcone derivative group, i.e., $\delta = 30$ mg cm⁻³. s.d. of BMD from previous studies was 15 mg/cm³, i.e., $S = 15$ mg cm⁻³. We selected the significant level at 5% ($\alpha = 0.05$) in a two-tailed test and power of the study at 90% (1–$\beta = 0.9$). According to the formula: $n = 2 [(U_\alpha + U_\beta) \, S \, \delta^{-1}]^2$, $U_{0.05} = 1.960$, $\beta = 0.10$, $U_{0.1} = 1.282$, the minimum sample size could be 5.3 in each group. We chose $n = 9$ for each group, which was enough to detect the real difference among the treatment groups. Mice in poor body condition, such as tumors or other sick conditions during aging, were excluded. The exclusion was made before random group assignments, experimental intervention and data analysis.

**Data availability.** The authors declare that all data supporting the findings of this study are available within the article and its Supplementary Information files or are available from the authors upon reasonable request.

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

## Acknowledgements

We thank technical staffs (Ms. Yeuk Siu Cheung and Mr. Chi Leung Chan) from Law Sau Fai Institute for Advancing Translational Medicine in Bone and Joint Diseases, School of Chinese Medicine, Hong Kong Baptist University for providing technical support. This work was supported by the Ministry of Science and Technology of China (2013ZX09301307 to A.L.), the Hong Kong General Research Fund (HKBU12100918 to G.Z., HKBU479111 to G.Z., HKBU478312 to G.Z., HKBU262913 to G.Z., HKBU12102914 to G.Z., HKBU261113 to A.L., CUHK14112915 to B.-T.Z., and CUHK489213 to B.-T.Z.), the Natural Science Foundation Council of China (81272045 to G.Z., 81700780 to C. L, 81371989 to S.P., 81272045 to B.G., 81401833 to B.G., and 81470072 to X.H.), the Research Grants Council and Natural Science Foundation Council of China (N_HKBU435/12 to G.Z.), the Croucher Foundation (Gnt#CAS14BU/CAS14201 to A.L.), the Interdisciplinary Research Matching Scheme (IRMS) of Hong Kong Baptist University (RC-IRMS/12-13/02 to A.L. and RC-IRMS/13-14/02 to G.Z.), the Hong Kong Baptist University Strategic Development Fund (SDF13-1209-P01 to A.L. and SDF15-0324-P02(b) to A.L.), the Hong Kong Research Grants Council Early Career Scheme (489213 to G.Z.), the Inter-institutional Collaborative Research Scheme of Hong Kong Baptist University (RC-ICRS/14-15/01 to G.Z. and RC-ICRS/16-17/01 to A.L.), the Faculty Research Grant of Hong Kong Baptist University (FRG1/13-14/024 to G.Z., FRG2/13-14/006 to G.Z., and FRG2/14-15/010 to G.Z.), the China Academy of Chinese Medical Sciences (Z0252 and Z0293 to A.L.).

## Author contributions

G.Z., A.L., and B.-T.Z. supervised the whole project. C.L., S.P., J.L. J.Lu., D.G., and F.J. performed the major research and wrote the manuscript in equal contribution. C.Lu., F. L., X.H., D.W.T.A., H.Z., and D.Y. provided their professional expertize.

## Additional information

**Competing interests:** The authors declare no competing interests.

