## [Peer Review File · Nature Communications]

Reviewers' comments:

Reviewer #1 (Remarks to the Author):

This is an extensive study looking at confirmation of both local and systemic inhibition of smurf1 in osteoblasts showing anabolic bone formation and enhancement of bmp-2 activity. It is very well done and comprehensive. No substantive criticisms.

Reviewer #2 (Remarks to the Author):

In this manuscript, the authors describe a small molecule strategy to enhance bone formation via inhibiting BMP-SMAD ubiquitination in osteoporotic patients. They show that age-related osteoporotic patients displayed various BMP-2 levels and Smurf1 activities. Patients with a normal BMP-2 level and elevated Smurf1 activity displayed poor responsiveness to rhBMP-2 during spinal fusion, while patients with lower BMP-2 level and normal Smurf1 activity were more responsive to BMP2. The authors identified a mouse correlate to the human population based on changes in serum osteocalcin, which they correlated to the levels of BMP-2 and Smurf1 in ovariectomized aged mice. The authors identified a small molecule capable of inhibiting Smurf1 thereby increasing osteogenesis and bone formation in vivo using these mice for spinal fusion. The data presented are compelling and appropriate controls were used. The conclusion are well justified by the presented data.

Reviewer #3 (Remarks to the Author):

This article interestingly performed Smurf1-based molecular docking and identified a chalcone derivative, which effectively targeted Smurf1 for inhibition, increased BMP signaling and promoted in vitro osteogenic differentiation and in vivo local bone formation. The in vivo data is especially impressive. However, I like the authors to shed more light on the following:

1. What information is known about the requirement of both the WW1 and WW2 domains for the function of Smurf1. If the references 10, 23 and 24 does not adequately explain, please include specific references that describe WW1-WW2 domains forming the binding pocket together to interact with target protein.
2. The authors should also provide functional specificity of the chalcone derivative by showing lack of activity on TGFB pathway (although some evidence is shown on Smad2) by looking at TGFB target gene expression or reporter activity.

Reviewer 1's comments: This is an extensive study looking at confirmation of both local and systemic inhibition of smurf1 in osteoblasts showing anabolic bone formation and enhancement of bmp-2 activity. It is very well done and comprehensive. No substantive criticisms.

Our response: Thanks for the positive and encouraging comments.

Reviewer 2's comments: In this manuscript, the authors describe a small molecule strategy to enhance bone formation via inhibiting BMP-SMAD ubiquitination in osteoporotic patients. They show that age-related osteoporotic patients displayed various BMP-2 levels and Smurf1 activities. Patients with a normal BMP-2 level and elevated Smurf1 activity displayed poor responsiveness to rhBMP-2 during spinal fusion, while patients with lower BMP-2 level and normal Smurf1 activity were more responsive to BMP2. The authors identified a mouse correlate to the human population based on changes in serum osteocalcin, which they correlated to the levels of BMP-2 and Smurf1 in ovariectomized aged mice. The authors identified a small molecule capable of inhibiting Smurf1 thereby increasing osteogenesis and bone formation in vivo using these mice for spinal fusion. The data presented are compelling and appropriate controls were used. The conclusion is well justified by the presented data.

Our response: Thanks for the positive and encouraging comments.

Reviewer 3's comments: This article interestingly performed Smurf1-based molecular docking and identified a chalcone derivative, which effectively targeted Smurf1 for inhibition, increased BMP signaling and promoted in vitro osteogenic differentiation and in vivo local bone formation. The in vivo data is especially impressive. However, I like the authors to shed more light on the following:

1. What information is known about the requirement of both the WW1 and WW2 domains for the function of Smurf1. If the references 10, 23 and 24 does not adequately explain, please include specific references that describe WW1-WW2 domains forming the binding pocket together to interact with target proteins.
2. The authors should also provide functional specificity of the chalcone derivative by showing lack of activity on TGFB pathway (although some evidence is shown on Smad2) by looking at TGFB target gene expression or reporter activity.

Our response: Thanks for the encouraging and constructive comments. We made the following point-by-point responses in our revised manuscript:

1. Regarding the reviewer 3's comment 1, the previous reports suggest that Smurf1 uses a coupled WW domain binding mechanism in its interaction with Smads (Aragon et al., 2011; Chong et al., 2010). The WW2 domain mainly responsible for the canonical binding with PY motif in Smads while the WW1 domain contacts the phosphorylation regions in linker of Smads (Aragon et al., 2011). The coupled WW domains enhance the affinity of Smurf1 with Smads when compared to the single WW2 domain (Chong et al., 2010). Inhibition of either WW1 or WW2 domain could decrease the interaction between Smurf1 and Smads (Cao et al., 2014; Kato et al., 2011). The above studies demonstrate that WW1 and WW2 domains cooperate to maintain the functions of Smurf1. In our revised manuscript, we supplemented additional references (Chong et al., 2010; Kato et al., 2011) to explain the requirement of both the WW1 and WW2 domains for the interaction between Smurf1 and target proteins. **Please refer to the highlighted revision (line number: 201-204, 730-732 and 737-739) in "Results" and "References".**

2. Regarding the reviewer 3's comment 2, it has been reported that transforming growth factor- β s (TGF- β s) and BMPs belong to the same TGF- β superfamily (Chen and Ten Dijke, 2016; Wu et al., 2016). TGF- β s activate canonical Smad2/3 signaling as well as non-canonical MAPK pathways (such as ERK1/2 and p38) to regulate numerous cellular processes (Chen et al., 2012; Chen and Ten Dijke, 2016; Smith et al., 2012). In this revised manuscript, we examined whether the chalcone derivative had effects on downstream molecules of TGF- β s including Smad2/3, ERK1/2 and p38. We found that the chalcone derivative didn't affect the expression and activation (phosphorylation) of the above molecules, suggesting the high specificity of the chalcone derivative on Smurf1 inhibition and BMP signaling. **Please refer to Supplementary Fig. 8f and the highlighted revision (line number: 218-223 and 552-563) in "Results" and "Methods".**

References

1. Aragon, E., Goerner, N., Zaromytidou, A.I., Xi, Q., Escobedo, A., Massague, J., and Macias, M.J. (2011). A Smad action turnover switch operated by WW domain readers of a phosphoserine code. *Genes Dev* 25, 1275-1288.
2. Cao, Y., Wang, C., Zhang, X., Xing, G., Lu, K., Gu, Y., He, F., and Zhang, L. (2014). Selective small molecule compounds increase BMP-2 responsiveness by inhibiting Smurf1-mediated Smad1/5 degradation. *Scientific reports* 4, 4965.
3. Chen, G., Deng, C., and Li, Y.P. (2012). TGF-beta and BMP signaling in osteoblast differentiation and bone formation. *International journal of biological sciences* 8, 272-288.
4. Chen, W., and Ten Dijke, P. (2016). Immunoregulation by members of the TGFbeta superfamily. *Nature reviews Immunology* 16, 723-740.
5. Chong, P.A., Lin, H., Wrana, J.L., and Forman-Kay, J.D. (2010). Coupling of tandem Smad ubiquitination regulatory factor (Smurf) WW domains modulates target specificity. *Proceedings of the National Academy of Sciences of the United States of America* 107, 18404-18409.
6. Kato, S., Sangadala, S., Tomita, K., Titus, L., and Boden, S.D. (2011). A synthetic compound that potentiates bone morphogenetic protein-2-induced transdifferentiation of myoblasts into the osteoblastic phenotype. *Mol Cell Biochem* 349, 97-106.
7. Smith, A.L., Robin, T.P., and Ford, H.L. (2012). Molecular pathways: targeting the TGF-beta pathway for cancer therapy. *Clin Cancer Res* 18, 4514-4521.
8. Wu, M., Chen, G., and Li, Y.P. (2016). TGF-beta and BMP signaling in osteoblast, skeletal development, and bone formation, homeostasis and disease. *Bone research* 4, 16009.

REVIEWERS' COMMENTS:

Reviewer #3 (Remarks to the Author):

The authors have adequately addressed the minor issues of the manuscript in its revised form. Overall, this work is impressive with strong in vivo data. Now, I recommend the manuscript to be accepted for publication.

Best